# OpenReview forum: "An Exploration of Non-Euclidean Gradient Descent: Muon and its Many Variants"
_ICML.cc/2026/Conference — ICML 2026 regular_

### Official Review · Reviewer_pfhj · 2026-02-15

**Soundness:** 3
**Presentation:** 4
**Significance:** 4
**Originality:** 2
**Overall Recommendation:** 5
**Confidence:** 4

**Summary:**

This paper studies the Muon optimizer and its practical variants through the lens of non-Euclidean steepest descent on neural network parameter spaces. The authors introduce a unifying framework in which (i) each layer has its own norm, (ii) an outer/product norm aggregates layers, and (iii) updates may be normalized or regularized. Within this framework, common practical recipes such as Muon+Adam (“MuonAdam”) are formalized, and new variants are proposed, most notably MuonMax and MuonMax-Momo. The paper also generalizes the Momo model-truncation step-size mechanism to arbitrary norms, arguing that it significantly improves robustness to learning-rate tuning.

**Compliance With Llm Reviewing Policy:**

Affirmed.

**Key Questions For Authors:**

how sensitive are the hyperparameters

**Limitations:**

Theoretical Guarantees Are Limited

**Strengths And Weaknesses:**

Overall, the authors focus on a relevant challenge: how to place Muon and related hybrid optimizers on a cleaner non-Euclidean steepest-descent footing, and how to turn that perspective into variants that are less brittle in practice. The research's core contribution consists of three parts: (1) a unified product-norm framework that formalizes practical MuonAdam-style updates as steepest descent on the full parameter space, rather than only at the layer level; (2) a generalization of Momo/model truncation to arbitrary norms, yielding Muon/Momo hybrids; and (3) a new optimizer, MuonMax, together with a stale nuclear-norm approximation intended to preserve most of the practical benefit without a major overhead. Empirically, the paper is strong: it evaluates a fairly broad family of variants, includes both language-modeling and image-classification experiments, and the main message is consistent across plots and ablations—namely, that MuonMax-Momo and MuonAdam-Momo often match or outperform MuonAdam/Scion while being substantially more robust to learning-rate misspecification. I also found the conceptual unification valuable, because it clarifies what exactly the practical MuonAdam implementation is optimizing and exposes previously unexplored design choices.

That said, I also see several limitations. First, the theoretical contribution is mainly a reformulation/unification plus closed-form derivations; it does not yet provide a deeper convergence or stability theory specifically explaining why MuonMax or MuonMax-Momo should outperform or be more robust than prior variants. next, although the experiments are systematic, the largest headline language-model results in the main body are still at moderate scale, and it remains unclear how much of the advantage persists in longer-horizon or truly frontier-scale training regimes. Then, the stale nuclear-norm approximation looks effective empirically, but the justification is heuristic; I would have liked a sharper characterization of when this approximation is safe or when it may fail. Finally, some of the empirical gains seem to come from robustness rather than clear best-case performance improvements, so the paper would benefit from a more explicit discussion of the tradeoff between “best achievable loss after heavy tuning” and “performance under realistic tuning budgets.”

---

> ### Author Rebuttal · Authors · 2026-03-30
>
> Thank you for such a careful reading of our paper. I don't think we could have summarized the paper better than you did in your review! Your questions are also very pertinent.
>
> **Q1**. The theoretical contribution does not provide convergence or stability theory explaining why any variant should be better than another.
>
> **A1**. Yes, you are correct here. This is closely tied to the major open problem of providing a theory that could shown benefits of Muon vs Adam vs SGD, and we do not claim to resolve this problem in our submission. Instead, we aim to cast light on the question through extensive numerical exploration. The only other thing we can point to, is that given we do cast all methods as different non-Euclidean gradient methods, this opens the door for importing theory for non-Euclidean gradient methods (though not for Adam variants), such as [Kovalev25b] for instance. Though we provide no such convergence in this submission.
>
> [Kovalev25b] Non-Euclidean SGD for Structured Optimization: Unified Analysis and Improved Rates, Dmitry Kovalev, Ekaterina Borodich,  arXiv:2511.11466
>
> **Q2**. Experiments are moderate scale. It is unclear if the advantage persists in frontier-scale regimes.
>
> **A2**. To scale up the experiments from the submission, we have added an evaluation of the GPT2-Large model (774M params) for SlimPajama with 15B tokens, comparing MuonAdam and MuonMax-Momo. These runs use about 20 tokens/parameter as recommended by the Chinchilla scaling laws. The results here are very similar to those of the SlimPajama6B experiments from our submission: MuonAdam and MuonMax-Momo achieve about the same loss when both are tuned, but MuonMax-Momo is much more robust to the choice of learning rate, and achieves a lower loss than MuonAdam for three out of four learning rates we tried. The results can be seen in Figure 1 of [this document](https://anonymous.4open.science/r/26_icml_rebuttal-FC40/rebuttal.pdf).
>
> Though this isn't anywhere near close to the frontier-scale, this is the most we can afford in terms of compute. We will however write an additional paragraph on how to use a scaling law to set a target loss, that could then be used in conjunction with our momo variants for training larger models.
>
> **Q3**. Justification of stale nuclear norm approximation is heuristic.
>
> **A3**. Yes, you are also correct here. We could add an initial discussion on how one could use Lipschitz continuity of the gradients to argue that the dual norm from one iteration to the next should change at a rate directly proportional to the learning rate. Thus, for smaller learning rates this stale approximation is reasonable, and it becomes riskier at larger learning rates. We can also argue, though not completely rigously, how the momo step sizes can help to guard against an inaccurate stale nuclear norm by ensuring that no updates are too large. Though none of the above would truly constitute a sharp characterization of when this approximation is safe.
>
> **Q4**. The paper would benefit from a more explicit discussion of the tradeoff between "best achievable loss after tuning" and "performance under realistic tuning budgets".
>
> **A4**. Good point, and this discussion becomes even more interesting when it comes to budgeting for training a truly large scale model, where such discussions are necessary. Where-in, part of the total budget would be used to form a scaling law and build a power law for the training loss, and the remaining budget for training the larger model with the extrapolated hyper-parameters and number of tokens. In such a setting, using a momo variant of the optimizer would come at no additional cost, since the fitted power law already provides an estimate of the attainable loss. We will add such a discussion. Is this what you had in mind, or are you thinking of another such trade-off?

---

> > ### Author Rebuttal · Reviewer_pfhj · 2026-03-31
> >
> > The authors have added the GPT experiment and acknoledged all my questions/concerns. I have no further questions to add and congrats again on the great paper

---

> > > ### Author Response · Authors · 2026-04-07
> > >
> > > Thanks for your kind words! Your suggestions definitely helped us improve the paper.

---

### Official Review · Reviewer_hVwe · 2026-02-18

**Soundness:** 4
**Presentation:** 4
**Significance:** 3
**Originality:** 2
**Overall Recommendation:** 5
**Confidence:** 4

**Summary:**

The authors propose a unified framework for non-Euclidean steepest descent to analyze the recently proposed Muon optimizer and its variants. They categorize existing methods, such as MuonAdam and Scion, as instances of Constrained Steepest Descent (CSD) defined by specific layer-wise and outer norms. The paper introduces a new variant called MuonMax, which is formulated as Regularized Steepest Descent (RSD). Furthermore, the authors integrate these non-Euclidean methods with Model-based Momentum (Momo) to address the known sensitivity of Muon to learning rate tuning. Extensive experiments on language modeling (GPT-2, Qwen2-MoE) and image classification (CIFAR10 and CIFAR100) demonstrate that the proposed variants, particularly MuonMax-Momo, significantly improve robustness to hyperparameter selection compared to standard baselines.

**Compliance With Llm Reviewing Policy:**

Affirmed.

**Key Questions For Authors:**

- In Figure 3, MuonAdam-Momo (CSD) performs comparably to MuonMax-Momo (RSD) in terms of final loss and robustness. What is the theoretical or practical advantage of switching to the Regularized formulation (MuonMax) if the Constrained formulation (MuonAdam) combined with Momo already solves the robustness issue?

- Regarding the Model-based Momentum hyperparameter F*: Your sensitivity analysis suggests it is robust for the tested tasks. How does the optimizer behave in tasks where the loss lower bound is unknown or significantly greater than zero (e.g., noisy regression)? Does the difficulty of estimating F* in such cases trade one tuning problem for another?

- The proposed method requires computing an outer norm that aggregates values across all layers. While you show minimal overhead for GPT-2 sizes using stale approximations, does this global aggregation introduce synchronization barriers or communication bottlenecks in massive distributed training setups compared to layer-wise independent updates?

**Limitations:**

The authors adequately discuss limitations regarding the computational overhead of the nuclear norm and the sensitivity to the new hyperparameter F*. They propose stale approximations to mitigate the computational cost.

**Strengths And Weaknesses:**

__Strengths:__
The paper provides a solid theoretical contribution by formalizing ad-hoc optimizers like Muon and Scion into a coherent framework of steepest descent. The distinction between Layer Norms (geometry) and Outer Norms (coupling) offers a clear taxonomy for understanding these algorithms. The integration of Momo is a practical and effective solution to the primary weakness of Muon, which is its brittleness regarding learning rate selection. The empirical evaluation is thorough; the authors test across different domains (NLP and Vision) and model architectures.

__Weaknesses:__
While the proposed MuonMax-Momo is presented as the superior method, the experimental results indicate that MuonAdam-Momo (simply applying Momo to the existing CSD formulation) also achieves excellent robustness and performance. The paper does not fully justify why the RSD formulation (MuonMax) is strictly necessary if the CSD formulation with Momo performs similarly. Furthermore, while Momo reduces sensitivity to the learning rate, it relies on the loss lower bound estimate F*. The analysis of F* is present but could be expanded to cover cases with high irreducible error or unknown lower bounds. Finally, the dependence on a global outer norm in MuonMax might introduce communication overheads in large-scale distributed training, which is not deeply explored beyond the wall-clock time measurements on smaller setups.

---

> ### Author Rebuttal · Authors · 2026-03-30
>
> Thank you for the feedback! We think that the paper will definitely be improved after incorporating answers to your questions.
>
> **Q1**. While the proposed MuonMax-Momo is presented as the superior method, the experimental results indicate that MuonAdam-Momo (simply applying Momo to the existing CSD formulation) also achieves excellent robustness and performance.
>
> **A1**. Thank you for pointing this out. This is probably an issue with our writing. It was not our intention to advertise MuonMax-Momo as the superior method, but instead, that using the "-Momo" variants leads to consistent improvement in robustness (and even best validation loss sometimes). We will make sure this point is clear in our revision.
>
> **Q2**. Justify why the RSD formulation (MuonMax) is strictly necessary if the CSD formulation with Momo performs similarly.
>
> **A2**. This is an interesting framing. We did not know a priori which variant RSD (MuonMax) or CSD (Momo) would perform better. We reported all results to share with the community how the choice of product norm and CSD/RSD affects the performance. That being said, we did find that RSD (MuonMax) is sometimes more robust, e.g. Figure 1, so neither of our proposed algorithms dominates the other in all cases.
>
> Also, it's interesting that a regularized method (RSD) can be superior when normalization (CSD) is the standard for deep learning. So we wanted to share with the empirical community that regularized methods can work well, and that this is not such a critical design choice in practice. This is important for theoreticians, who consider mostly only the theory of CSD (e.g. Muon) because it was considered to be more practical, despite being harder to analyze. Indeed, RSD methods (e.g. MuonMax) are easier to analyze, since they are the generalization of standard gradient descent (as oppossed to Frank-Wolfe or conditional gradient descent).
>
> **Q3**. How do the Momo optimizers behave in tasks where the loss lower bound is unknown or significantly greater than zero? Are we just trading tuning one hyperparameter for another?
>
> **A3**. Good question. First, in the setting where there is no known lower bound. This is a slightly less common setting, but it does occur, for instance when minimizing the ELBO in variational inference. In such settings we would recommend either a) Do not use the momo variants or b) build a scaling lower from training smaller sized problem instances, fit a power law of the minimal achievable loss, then use this power law (with some buffer) as the lower bound of a larger model instance. We will add a discussion regarding using scaling laws to set a lower bound for such cases.
>
> As for the setting where the known lower bound is significantly greater than zero, we would argue that language modeling is exactly such a setting. The smallest achievable loss is often around 2 for bigger models, and 3 for smaller models, both of which are far from zero (at least when compared to image classification losses). It was in this setting where we show that using loss lower bound of zero still works  well (see Figure 4). This figure makes it clear that there is little sensitivity to the chosen lower bound, certainly far less sensitivity than for the learning rate with say MuonAdam, so the need to choose $F_*$ is a much smaller burden than the need to choose the learning rate.
>
> **Q4**. Does the global aggregation of nuclear norms introduce synchronization errors or communication bottlenecks in massive distributed training setups compared to layer-wise independent updates?
>
> **A4**. Great question, and we believe that the answer is no, as long as you use stale nuclear norms. With stale nuclear norms, the dependencies between layers is exactly the same as in layer-wise independent updates (like Muon): at each iteration, every layer only needs its own momentum buffer and other information (e.g. current parameters, stale nuclear norms) which are computed and shared during the previous iteration. Also, the cost to share the nuclear norms between layers is tiny: there is no added latency since we can do this at a pre-existing synchronization point, and there is negligible added bandwidth cost because we only have to communicate one scalar value for each layer. Lastly, the concern only applies in the case of pipeline parallelism; under data parallelism and tensor parallelism, every device holds a full copy of all parameters and their momentum buffers, so no synchronization is required even to share the non-stale nuclear norms. We can add this discussion to the revised version of the paper.

---

> > ### Author Rebuttal · Reviewer_hVwe · 2026-03-31
> >
> > I appreciate the authors’ effort in providing a clear and thoughtful rebuttal.
> > The rebuttal addresses most of my concerns clearly and professionally. I appreciate the clarification that MuonMax-Momo is not intended to strictly dominate MuonAdam-Momo, but rather that the main message is the consistent robustness gain from the Momo variants. This clarification is consistent with the empirical results.
> >
> > I also find the response on the distributed overhead question reasonable within the scope of the paper, especially the explanation based on stale nuclear norms. The discussion of the lower-bound parameter $F_{*}$ is helpful as well, although I still think the paper would benefit from a clearer statement of scope for settings where a reliable lower bound is not available.
> >
> > One point that would still strengthen the paper is a more explicit characterization of when one should prefer the RSD formulation over the CSD formulation with Momo. The rebuttal makes clear that MuonMax-Momo and MuonAdam-Momo are both strong, but the paper still stops short of explaining the practical regimes in which MuonMax-Momo is preferable. For example, is its advantage mainly visible at larger scale, under greater learning-rate misspecification, or under certain matrix/non-matrix parameter ratios?
> >
> > Overall, my main concerns are largely addressed.

---

> > > ### Author Response · Authors · 2026-04-07
> > >
> > > Thank you for your positive comments. On the point about RSD vs. CSD, it really is unclear when one should be preferred to the other. It seems that the superior method changes depending on the dataset and architecture in ways that are somewhat opaque, but after all that is deep learning. One concrete thing we can say is that RSD tends to be inferior unless we also use model truncation, which aligns with the common idea in deep learning that updates should have some form of normalization or clipping. Interestingly, even without model truncation some RSD variants still outperform Adam, e.g. 3.791 final loss of MuonMax (Table 2) compared to ~3.75 final loss of Adam in the same setting (Figure 7).

---

### Official Review · Reviewer_dSBX · 2026-03-08

**Soundness:** 1
**Presentation:** 2
**Significance:** 2
**Originality:** 2
**Overall Recommendation:** 2
**Confidence:** 3

**Summary:**

The paper sets up a general mathematical framework that includes many of the Muon family algorithms. According to the framework, the differences between these variants are basically in the different choice of layer norms and the way they are aggregated. The paper also proposed a new variant under this framework. Empirical results show that different variants have different sensitivity on the learning rate choices.

**Compliance With Llm Reviewing Policy:**

Affirmed.

**Final Justification:**

I have read the rebuttal and discussed with other reviewers. My original concerns are partially addressed but mostly remain, and I don't see the contribution is significant enough for ICML.

**Key Questions For Authors:**

N/A

**Limitations:**

yes

**Strengths And Weaknesses:**

[S1] The proposed algorithm improves robustness to learning rate tuning.

[W1] Most of the effort of the work is to set up this general mathematical framework to include different designs. However, it is unclear what is the benefit of doing this generalization. Does it provide a deeper understanding of these algorithms? Does it facilitate any better theoretical guarantees? Based on what has been presented in the paper, I do not see any of these, and I cannot agree with the claim (it) “strengthen the theoretical foundation of MuonAdam”.

[W2] The paper also proposed a new variant of the Muon family. However, a justification of why this variant is better than others is absent. It seems like it is just another way of choosing the layer norms and norm aggregation under this framework. But what makes this choice better than others? No intuition discussed, no theory derived.

[W3] The proposed algorithm does not improve the validation performance in the setting of optimal learning rate.

---

> ### Author Rebuttal · Authors · 2026-03-30
>
> Thank you for your review and constructive feedback. Below we responded to your individual points.
>
> **Q1**. It is unclear what is the benefit of setting up this mathematical framework. Does it provide a deeper understanding of these algorithms? Does it facilitate any better theoretical guarantees? Based on what has been presented in the paper, I do not see any of these, and I cannot agree with the claim (it) “strengthen the theoretical foundation of MuonAdam”.
>
> **A1**. We agree that our original wording “strengthen the theoretical foundation of MuonAdam” was too broad, and we will revise it to a more precise claim.
>
> Our theoretical framing has two immediate benefits: 1) It shows that there are many more variants of Muon than were previously considered and 2) it casts both new and existing variants such as MuonAdam, formally as a non-Euclidean descent method under an appropriate norm. Based on this framing, MuonAdam is just a version of non-Euclidean gradient descent. This is what we meant by "strengthen the theoretical foundation of MuonAdam", which we will now re-write as "we show that MuonAdam is a non-Euclidean steepest descent method."
>
> Furthermore, as a community we know a lot about gradient descent, so our framing enables the transfer of theoretical and algorithmic tools to MuonAdam and all the variants in our framework. One such theoretical tool is the analysis in [Kovalev25b], though this analysis does not hold specifically for MuonAdam, because of the Adam component. At this stage, the paper primarily introduces the conceptual framework and presents empirical findings; a comprehensive theoretical exploration is left for future work.
>
> On the practical side, we developed the first non-Euclidean model truncation which resulted in these "momo" variants of each method. Because we have framed MomoAdam, and all variants as a non-Euclidean descent method, we can now make use of model truncation to improve sensitivity of the learning for all these variants of Muon. Which is exactly what our experiments show.
>
> [Kovalev25b] Non-Euclidean SGD for Structured Optimization: Unified Analysis and Improved Rates, Dmitry Kovalev, Ekaterina Borodich,  arXiv:2511.11466
>
> **Q2**. The paper also proposed a new variant of the Muon family. However, a justification of why this variant is better than others is absent.
>
> **A2**. We agree that the motivation for the new variant should be explained more clearly, and we will improve this in the revision.
>
> MuonMax originated from the idea of changing MuonAdam's constrained update step on the product of all weight matrices to a regularized update step, while preserving the Adam update on all other parameters. We then found an outer norm that induces this kind of update, which is exactly the outer norm defined in Equation 23 which we use to define MuonMax.
>
> This explains where the algorithm came from, but we acknowledge that we do not a have a clear principle for favoring one variant over another. We agree that a deeper justification of one norm over others is an important problem, though this is a major open question for this whole research topic. If we could predict a priori which norm is best for steepest descent, we could potentially explain why Muon outperforms Adam and SGD. We do not claim to solve this major problem in our submission, rather we want to describe all variants under one formalism (non-Euclidean gradient), numerically explore the many variants and share the results with the community.
>
> We will revise the paper to include our intuition for MuonMax explicitly, since the current draft does not explain our design choice clearly enough.
>
> **Q3**. The proposed algorithm does not improve the validation performance in the setting of optimal learning rate.
>
> **A3**. This is a fair point, and we agree that our contribution is not to achieve strictly better performance after tuning.
>
> The main goal of our proposed algorithms is to improve robustness to learning rate choice, which is fundamental for large-scale settings where extensive tuning is expensive or impossible. Our results show that we consistently achieve this improved robustness, for example on SlimPajama1B (Figure 1), MuonMax-Momo achieves 3.25 loss or smaller for LRs across five orders of magnitude, while MuonAdam and Scion achieve the same only for a single LR. Similar behavior is observed in the 6B setting and in image classification experiments.
>
> Further, even after tuning, our new variants are always competitive with MuonAdam in terms of validation loss, and they are sometimes better, such as on SlimPajama1B (Figure 1) and CIFAR-100 (Figure 12).
>
> We will revise the paper to make our contribution more explicit: our methods are designed to reduce sensitivity to tuning, not to achieve uniformly better performance in all settings after exhaustive tuning.

---

> > ### Author Rebuttal · Reviewer_dSBX · 2026-04-04
> >
> > I thank the authors for the detailed response. My concerns are partially addressed, but mostly remain.
> >
> > W1: this framework does not strength the theoretical foundation. It seems just one way of putting several similar algorithms into one box. Still could not see the benefits of this "unification".
> >
> > W2: The paper also cannot predict/explain which variant is principly better over the others. The proposed variant is also not clearly better than others numerically.
> >
> > W3: The new variant does not improve performance in the optimal learning rate setting. While it is numerically more robust to bad learning rate choices, I don't think this is a contribution that is significant enough for an ICML publication.

---

> > > ### Author Response · Authors · 2026-04-07
> > >
> > > Thank you for continuing the conversation with us.
> > >
> > > **W1**. As we stated in our rebuttal, we agree that the wording "we strengthen the theoretical foundation" is too broad and imprecise, which we will rewrite as "we show that MuonAdam is a non-Euclidean steepest descent method". As for the benefit of the theoretical framework: the framework is exactly what enables the derivation of the many variants we explored in the paper, as well as the use of Momo stepsizes for these variants. We emphasize that one cannot extend Momo for MuonAdam without first writing MuonAdam as a genuine steepest descent method on the space of all parameters.
> > >
> > > **W2**. We acknowledge that we cannot predict which variant will be better than others, and we reiterate that this is actually a major open problem for the Muon-adjacent research area. Why is optimizing w.r.t. the spectral norm (Muon) better than w.r.t. the Euclidean norm (SGD) or the $\ell_\infty$ norm (signSGD)? When will optimizing w.r.t. one norm be better than optimizing w.r.t. another? We don't know the answer to these questions. They are central questions in this line of research, but we make no claim of answering them in this paper. We do not agree that not solving a major open problem should disqualify a paper from publication.
> > >
> > > Also, we disagree that the proposed methods are not clearly better than the baselines in experiments. Our Figure 1 shows that with 1B SlimPajama tokens, MuonMax-Momo achieves a lower loss than both MuonAdam and Scion for every single learning rate in the grid. Very similar behavior is observed as we scale up to 6B and 15B tokens. In other cases, even when our methods don't perform uniformly better for every single learning rate, the robustness to choice of learning rate (as measured by LR basin width) is consistently improved by a large margin compared to baselines.
> > >
> > > **W3**. We agree that the proposed methods do not always improve performance after tuning, and we reiterate that our goal is to reduce the need for tuning, rather than to improve performance after tuning. The reviewer says that robustness to the choice of learning rate is not a contribution significant enough for ICML, but we disagree: there is already an established body of work at top-tier ML venues focused on reducing the need for LR tuning, such as with Polyak-type stepsizes [1, 2, 3], parameter-free methods [4, 5, 6], schedule-free optimization [7], and learning rate transfer [8]. These methods are useful because in practice one often does not know optimal hyperparameters a priori and one cannot always tune excessively to find them, especially at large scale. In the same vein, our methods provide value to the community by achieving the performance of Muon (or better) while being much easier to tune.
> > >
> > > [1] Loizou, Nicolas, et al. "Stochastic polyak step-size for sgd: An adaptive learning rate for fast convergence." International conference on artificial intelligence and statistics. PMLR, 2021.
> > >
> > > [2] Orvieto, Antonio, Simon Lacoste-Julien, and Nicolas Loizou. "Dynamics of sgd with stochastic polyak stepsizes: Truly adaptive variants and convergence to exact solution." Advances in Neural Information Processing Systems 35 (2022): 26943-26954.
> > >
> > > [3] Schaipp, Fabian, Robert M. Gower, and Michael Ulbrich. "A Stochastic Proximal Polyak Step Size." Transactions on Machine Learning Research (2023).
> > >
> > > [4] Defazio, Aaron, and Konstantin Mishchenko. "Learning-rate-free learning by d-adaptation." International conference on machine learning. PMLR, 2023.
> > >
> > > [5] Mishchenko, Konstantin, and Aaron Defazio. "Prodigy: An Expeditiously Adaptive Parameter-Free Learner." International Conference on Machine Learning. PMLR, 2024.
> > >
> > > [6] Ivgi, Maor, Oliver Hinder, and Yair Carmon. "DoG is SGD’s best friend: A parameter-free dynamic step size schedule." International conference on machine learning. PMLR, 2023.
> > >
> > > [7] Defazio, Aaron, et al. "The road less scheduled." Advances in Neural Information Processing Systems 37 (2024): 9974-10007.
> > >
> > > [8] Yang, Greg, et al. "Tensor Programs V: Tuning large neural networks via zero-shot hyperparameter transfer." Advances in Neural Information Processing Systems 34 (2021): 17084-17097.

---

### Official Review · Reviewer_4Sbu · 2026-03-13

**Soundness:** 3
**Presentation:** 3
**Significance:** 2
**Originality:** 3
**Overall Recommendation:** 5
**Confidence:** 3

**Summary:**

The paper unifies Muon, Adam, and several variants under a single steepest descent framework. Instead of treating Muon as a standalone trick, they show it is a point in a design space with three independent axes:
1. Constrained vs regularized steepest descent
2. Outer norm choice for aggregating across layers
3. Per-parameter norm choice
They show how this framework can incorporate techniques such as Momo, which selects an adaptive step size.
This paper systematically investigates empirical performance under these choices and demonstrate that MuonMax variant achieves strong performance on language modeling across a range of learning rates.

**Compliance With Llm Reviewing Policy:**

Affirmed.

**Final Justification:**

The strengths of presenting a unifying framework and algorithms which are robust across their ranges outweigh the downsides (e.g., lack of predictive theory). We should be designing algorithms which are robust rather than purely focusing on the best possible result.

**Key Questions For Authors:**

n/a

**Limitations:**

The authors do not explicitly describe their limitations and could strengthen the paper by acknowledging limitations and directions for future work.

**Strengths And Weaknesses:**

Strengths:

- The steepest descent framework is principles and captures a variety of existing algorithms. The framework is useful for extending and contextualizing a recent popular algorithm for training language models.
- Empirically, the design of the experiment is sound and demonstrates the significance of the results. The authors conduct a grid search across four different design choices (constrained vs regularized steepest descent, outer norm, additional norm, model truncation) to arrive at their conclusions.
- The learning rate basin metric suggests that the proposed variants are less sensitive to changing learning rates.
- The paper is clear to follow.

Weaknesses.
- The experiments are conducted purely in the context of training large language models. While this is an area of interest, it would improve the significance of these results if other domains are also considered. While there are experiments with ResNets in the appendix, I am concerned that the constant learning rate schedule results in a bias against the baseline approaches.

---

> ### Author Rebuttal · Authors · 2026-03-30
>
> Thank you for your efforts in the reviewing process, and for your helpful suggestions. Below we responded individually to your points.
>
> **Q1**. While there are experiments with ResNets in the appendix, I am concerned that the constant learning rate schedule results in a bias against the baseline approaches.
>
> **A1**. Thank you for the suggestion, we have added image classification experiments with a cosine LR schedule, and the results can be seen in Figures 2 and 3 of [this document](https://anonymous.4open.science/r/26_icml_rebuttal-FC40/rebuttal.pdf). For simplicity, we keep all aspects of training besides the LR schedule the same as in our original experiments.
>
> For CIFAR10, the results are basically consistent with those from the submission: MuonMax-Momo has the lowest training loss and highest validation accuracy for almost every learning rate. When the learning rates are tuned, MuonAdam and MuonAdam-Momo achieve a slightly lower loss than MuonMax-Momo.
>
> For CIFAR100, the methods have different behavior. In terms of training loss, MuonAdam is the best after tuning, and all methods have a similar sensitivity to the choice of LR. In terms of validation accuracy, Adam, MuonAdam, and MuonMax-Momo reach a similar score after tuning and again have similar LR sensitivity. Notably, both MuonAdam and MuonAdam-Momo diverge when the LR is one order of magnitude larger than the tuned value. Here it seems cosine schedule benefits more Adam than the Muon based methods. We also observe that Muon is often implemented with a constant and linear decay schedule instead, and this may be the reason why.
>
> **Q2**. The authors do not explicitly describe their limitations and could strengthen the paper by acknowledging limitations and directions for future work.
>
> **A2**. Thank you, this is a good suggestion. Often ICML gives accepted papers an additional page. We would use this page to write a clear and transparent limitations section, and future work. This will include open questions such as developing a theory that could guide the choice of product norm, and the development of a scaling law to set a more precise target loss for the momo variants.

---

> > ### Author Rebuttal · Reviewer_4Sbu · 2026-04-03
> >
> > I have read the other reviews and responses. I agree with the authors about the utility of placing these prior algorithms into a single framework and am glad they are clarifying the contribution as "we show that MuonAdam is a non-Euclidean steepest descent method" I think the CIFAR-100 results could use some further exploration to understand, but it is okay to defer as beyond the scope of this work.

---

> > > ### Author Response · Authors · 2026-04-07
> > >
> > > Thank you for your positive evaluation of our paper. We agree that the difference in behavior across CIFAR-10 vs. CIFAR-100 is interesting and requires some work to understand, and we will add a comment in the final version pointing this out.

---

### Decision · Program_Chairs · 2026-04-30

**Decision:**

Accept (regular)

**Comment:**

This paper sets up a framework that encompasses many algorithms in the Muon family, including MuonAdam. From the proposed framework, a new variant called MuonMax-MoMo is proposed, which is insensitive to the learning-rate parameter empirically. The proposed framework, in a nutshell, consists of defining a notion of an outer norm to aggregate the layer-wise norms, and the choice of the layer-wise norm and the way they are aggregated lead to different optimization updates.

While the proposed algorithm MuonMax-MoMo has the appealing feature of being empirically robust to the learning rate, one criticism raised by one of the reviewers is that there is no convergence-rate analysis, and a theoretical justification for why this variant is better than others, as well as guidance for deciding a choice in the proposed framework, are absent. There is no intuition discussed and no theory derived to explain why MuonMax-MoMo is insensitive to the choice of the learning rate.

For these concerns, there was an internal discussion where all the reviewers were involved and shared some thoughts. It appears that the majority of the reviewers agree on the strengths and weaknesses of this paper, while differing on how they weigh the strengths and weaknesses.

With that said, I believe the empirical findings showing that the proposed variant is insensitive to the learning-rate parameter is a merit worthy of ICML. However, whether the method is insensitive to other parameters is not crystal clear in its current form.

For all these reasons, this paper is recommended for a weak accept.